# Shared mushroom body circuits underlie visual and olfactory memories in *Drosophila*

Katrin Vogt[1†], Christopher Schnaitmann[1†‡], Kristina V Dylla[1§], Stephan Knapek[1], Yoshinori Aso[2], Gerald M Rubin[2], Hiromu Tanimoto[1,3*]

[1]Max-Planck-Institute of Neurobiology, Martinsried, Germany; [2]Janelia Farm Research Campus, Howard Hughes Medical Institute, Ashburn, United States; [3]Graduate School of Life Sciences, Tohoku University, Sendai, Japan

**Abstract** In nature, animals form memories associating reward or punishment with stimuli from different sensory modalities, such as smells and colors. It is unclear, however, how distinct sensory memories are processed in the brain. We established appetitive and aversive visual learning assays for *Drosophila* that are comparable to the widely used olfactory learning assays. These assays share critical features, such as reinforcing stimuli (sugar reward and electric shock punishment), and allow direct comparison of the cellular requirements for visual and olfactory memories. We found that the same subsets of dopamine neurons drive formation of both sensory memories. Furthermore, distinct yet partially overlapping subsets of mushroom body intrinsic neurons are required for visual and olfactory memories. Thus, our results suggest that distinct sensory memories are processed in a common brain center. Such centralization of related brain functions is an economical design that avoids the repetition of similar circuit motifs.

**\*For correspondence:**
hiromut@m.tohoku.ac.jp

†These authors contributed equally to this work

**Present address:** ‡Neurobiologie/Tierphysiologie, Institut für Biologie 1, Albert-Ludwigs-Universität Freiburg, Freiburg, Germany; §Department of Biology–Neurobiology, University of Konstanz, Konstanz, Germany

**Competing interests:** The authors declare that no competing interests exist.

**Reviewing editor**: Mani Ramaswami, Trinity College Dublin, Ireland

## Introduction

When animals encounter reward or harm, they form associations with concomitant environmental cues. Such associative memories allow an animal to predict upcoming events and to choose an appropriate behavior. Memory induced by appetitive and aversive events is usually not restricted to a single sensory cue. For example, a traumatic event drives aversive associative memories of concurrent auditory and visual stimuli in rats (*Campeau and Davis, 1995*). The same appetitive and aversive reinforcers drive both olfactory and visual memories in insects, while associative memories with different modalities are formed using the same neurotransmitter system (*Unoki et al., 2005*, *2006*). However, the circuit mechanisms underlying memories of different sensory modalities driven by the same reinforcing stimulus are unknown. Two alternative circuit organizations are possible: each sensory modality may feed into a dedicated memory circuit, or representations of different sensory stimuli (e.g., olfactory and visual) may undergo associative modulation in a shared set of neurons in the brain (*Zars, 2010*).

Cellular mechanisms underlying associative learning have been intensely studied in various animals, including *Drosophila* (*Keene and Waddell, 2007*). However, comparisons between memories of different sensory modalities have led to contradictory results. For example, the mushroom bodies (MBs) are required for olfactory and gustatory memories (*Heisenberg et al., 1985*; *Davis, 1993*; *Heisenberg, 2003*; *Masek and Scott, 2010*), but according to previous studies, not for a visual memory task (*Heisenberg et al., 1985*; *Wolf et al., 1998*; *Tang and Guo, 2001*; *Zhang et al., 2007*). Nevertheless, other studies suggest that visual information is indeed processed in the MBs (*Barth and Heisenberg, 1997*; *Liu et al., 1999*; *Brembs and Wiener, 2006*; *van Swinderen et al., 2009*). These discrepancies are difficult to resolve because many of these studies, especially those comparing stimuli with a different physical nature (e.g., olfactory vs visual), employed different behavioral tasks (e.g., flight orientation or binary choice by walking flies) and/or conditioning designs (*Brembs and Wiener, 2006*;

**eLife digest** Animals tend to associate good and bad things with certain visual scenes, smells and other kinds of sensory information. If we get food poisoning after eating a new food, for example, we tend to associate the taste and smell of the new food with feelings of illness. This is an example of a negative 'associative memory', and it can persist for months, even when we know that our sickness was not caused by the new food itself but by some foreign body that should not have been in the food. The same is true for positive associative memories.

It is known that many associative memories contain information from more than one of the senses. Our memory of a favorite food, for instance, includes its scent, color and texture, as well as its taste. However, little is known about the ways in which information from the different senses is processed in the brain. Does each sense have its own dedicated memory circuit, or do multiple senses converge to the same memory circuit?

A number of studies have used olfactory (smell) and visual stimuli to study the basic neuroscience that underpins associative memories in fruit flies. The olfactory experiments traditionally use sugar and electric shocks to induce positive and negative associations with various scents. However, the visual experiments use other methods to induce associations with colors. This means that it is difficult to combine and compare the results of olfactory and visual experiments.

Now, Vogt, Schnaitmann et al. have developed a transparent grid that can be used to administer electric shocks in visual experiments. This allows direct comparisons to be made between the neuronal processing of visual associative memories and the neural processing of olfactory associative memories.

Vogt, Schnaitmann et al. showed that both visual and olfactory stimuli are modulated in the same subset of dopamine neurons for positive associative memories. Similarly, another subset of dopamine neurons was found to drive negative memories of both the visual and olfactory stimuli. The work of Vogt, Schnaitmann et al. shows that associative memories are processed by a centralized circuit that receives both visual and olfactory inputs, thus reducing the number of memory circuits needed for such memories.

*Brembs and Plendl, 2008*; *Pitman et al., 2009*; *Ofstad et al., 2011*). We reasoned that a more informative comparison might be obtained using comparable learning paradigms (*Scherer et al., 2003*; *Gerber et al., 2004a*; *Guo and Guo, 2005*; *Hori et al., 2006*; *Mota et al., 2011*).

We previously developed appetitive and aversive visual conditioning assays (*Schnaitmann et al., 2010*). During appetitive training, flies receive one of the two color stimuli together with a sugar reward, whereas the other color is presented without a reward. When they are given the choice between the two colors in a subsequent test, flies show significant conditioned approach to the previously rewarded color. The same paradigm was also used with application of acid punishment during training instead of sugar reward, leading to conditioned avoidance. This visual conditioning assay, appetitive learning in particular, shares several critical features with the well-studied olfactory conditioning assay (*Tempel et al., 1983*; *Schwaerzel et al., 2003*), including the conditioning design, sugar-soaked filter paper as the reward, and the use of a binary choice between two conditioned stimuli scored as an alteration in the distribution of freely moving flies. Thus, our experimental design allows direct comparison of the mechanisms underlying appetitive visual memories with those of olfactory memories.

Studies that have found distinct neuromodulator circuits underlying appetitive and aversive memories for one modality have succeeded by restricting the critical difference to reward vs punishment (*Schwaerzel et al., 2003*; *Gerber et al., 2004a*; *Vergoz et al., 2007*; *Honjo and Furukubo-Tokunaga, 2009*; *von Essen et al., 2011*). However, there is no established aversive visual learning assay employing the widely used potent aversive reinforcer, electric shock, in the same way as in aversive olfactory conditioning (*Quinn et al., 1974*; *Tully and Quinn, 1985*). To meet this need, we implemented electric shock punishment into our learning assay by devising a transparent shock grid that is placed beneath the flies. This allows us to pair electric shock with the same visual stimuli as used in appetitive learning (*Schnaitmann et al., 2010*). Using these appetitive and aversive visual learning assays, we examined the roles of distinct aminergic neurons and found a common requirement of dopamine

neurons in visual and olfactory learning. Furthermore, we demonstrate a role for the MB for appetitive and aversive visual memories, suggesting significant commonality in the neuronal mechanisms underlying memories of different sensory modalities.

## Results

### Electric shock punishment induces aversive visual memories

We previously developed an appetitive visual learning assay that shares critical features with olfactory conditioning (*Figure 1A,B*; *Schnaitmann et al., 2010*). In our assay, the visual stimuli (LEDs; *Figure 1D*) are projected from below through translucent sugar-soaked filter paper, the appetitive reinforcer used in olfactory conditioning. However, the commonly used aversive reinforcer, electric shock, is more difficult to integrate, as a metal grid beneath the fly would disrupt visual stimulation from below that is used in appetitive conditioning.

We solved this problem by fabricating a shock grid from a transparent low-resistance material, indium tin-oxide (ITO; *Figure 1B,C*). An alternating electrode pattern was laser-etched into a thin layer of ITO on a glass plate (*Figure 1E*). Other parameters of the assay were replicated from appetitive conditioning, except that the arena height was reduced so that flies could not escape the electric shock.

To characterize shock punishment using the transparent grid, we subjected flies to visual conditioning with four training trials as for appetitive training (*Figure 1—figure supplement 1A*). During one training trial, each visual stimulus (green/blue color) was alternately presented for 1 min to the flies, one of them paired with punishment ('Materials and methods'). The electric shock served as potent aversive reinforcement and induced aversive visual memory at a signal to noise ratio comparable to visual memories in other paradigms (*Figure 1F*, *Figure 1—figure supplement 1A*; *Menne and Spatz, 1977*; *Wolf and Heisenberg, 1991*). We found that conditioned avoidance increased with ascending voltages. A plateau was reached at approximately 30 V, and the performance did not change with more intense shock (*Figure 1F*). Thus, we applied 60 V for all subsequent experiments. For aversive conditioning, a single shock pulse was applied 5 s before the beginning of the test to arouse the flies (*Figure 1—figure supplement 1B*; *Menne and Spatz, 1977*; *Gerber and Hendel, 2006*). Video analysis of the whole test period showed that the choice behavior stabilized within roughly 20 s (*Figure 1—figure supplement 1C*). Over 90 s of the test, flies' preference for a previously shock-paired visual stimulus is decreased in comparison to an unpaired visual stimulus (*Figure 1—figure supplement 1C*). Together with the previously developed appetitive memory assay, these behavioral tools allow us to compare the neural requirements of appetitive and aversive visual memory, as well as visual and olfactory memories.

### Different sets of dopamine neurons drive appetitive and aversive visual memories

Monoamine neurons were previously shown to signal reinforcement during olfactory memory formation in *Drosophila* (*Schwaerzel et al., 2003*; *Claridge-Chang et al., 2009*; *Aso et al., 2012*; *Burke et al., 2012*; *Liu et al., 2012*; *Sitaraman et al., 2012*) and other insects (*Hammer, 1993*; *Unoki et al., 2005*; *Vergoz et al., 2007*). In order to identify reinforcement signaling neurons for visual memories, we therefore blocked distinct sets of aminergic neurons by expressing *shibire*$^{ts1}$ (*shi*$^{ts1}$; *Kitamoto, 2001*) and assessed these neurons' role in appetitive and aversive conditioning. To target these aminergic neurons, we chose *TDC2-GAL4*, *TH-GAL4* and *DDC-GAL4* driver lines that label different subsets of tyramine/octopamine and dopamine neurons (*Li et al., 2000*; *Friggi-Grelin et al., 2003*; *Cole et al., 2005*). We found that the requirements of these neurons for appetitive and aversive visual memories are strikingly similar to those in olfactory memories. Blocking octopamine/tyramine neurons with *TDC2-GAL4* did not cause a significant defect in sucrose reward or shock punishment memory (*Figure 2A*). In contrast, the blockade of a large fraction of dopamine neurons with *TH-GAL4* selectively reduced aversive memory (*Figure 2A*). As in olfactory learning (*Liu et al., 2012*), the blockade with *DDC-GAL4* that labels different sets of dopamine and serotonin neurons substantially impaired appetitive memory, but not aversive memory (*Figure 2A*). The blockade with *DDC-GAL4* did not significantly affect the reflexive choice of sugar, while blocking the dopamine system with *TH-GAL4* caused prolonged hyperactivity that indirectly influenced shock avoidance (*Lebestky et al., 2009*; *Table 1*).

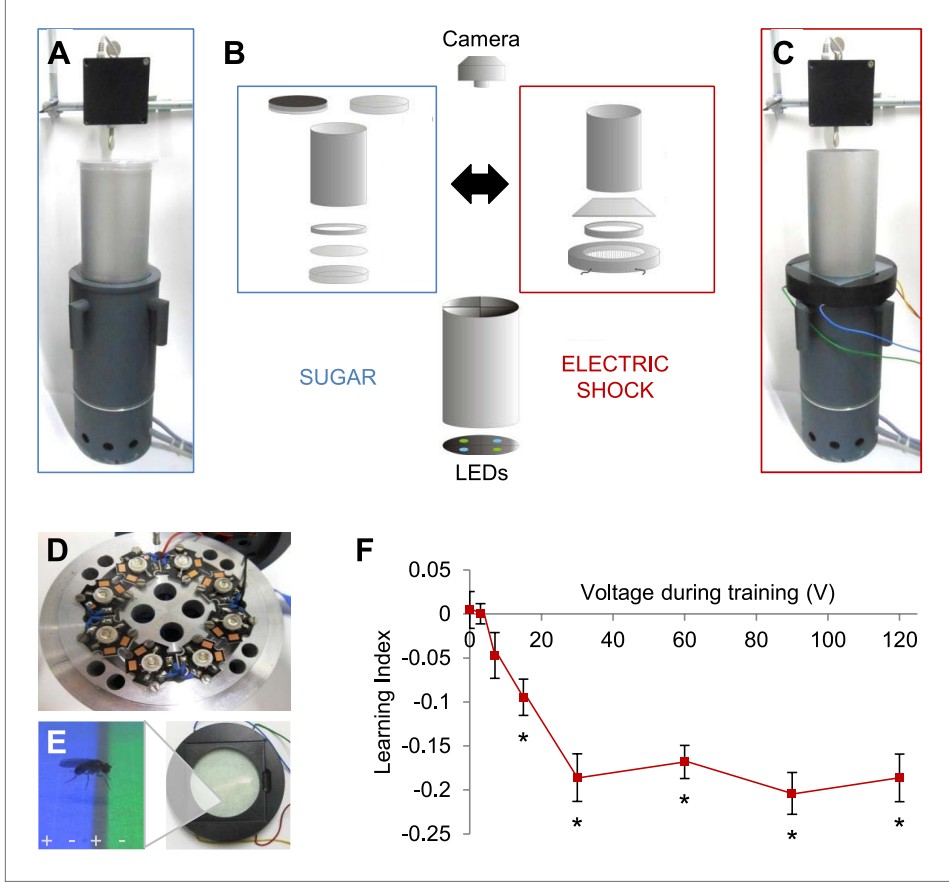

**Figure 1**. Modular appetitive and aversive visual learning. (**A**–**C**) Experimental setups for appetitive and aversive visual learning. Scheme shows single components (**B**) of exchangeable conditioning arenas for sugar reward (**A**) and electric shock punishment (**C**) that share the same light source and video camera (**B**). (**B**) Appetitive setup: cylindrical Fluon-coated arena closed from top with opaque lid during training or transparent lid during test. Exchangeable Petri dish on the bottom to present sugar or water soaked filter paper during training and neutral filter paper during test. Filter paper is clamped in the dish by a plastic ring. Aversive setup: the circular arena consists of a transparent electric shock grid, removable Fluon-coated plastic ring and transparent lid. The cylinder on top isolates each setup from the others and creates a similar closed visual scene as in the appetitive setup. (**D**) Visual stimulus source with one blue and one green high power LED per quadrant. (**E**) The conditioning arena with the transparent electric shock grid and a magnification with visual stimulation and a fly. Alternating stripes marked by + and − symbols indicate electric shock application. (**F**) Aversive visual memory depends on shock intensity (One-way ANOVA, $p < 0.001$). Flies show significant memory from 15 V (One sample $t$ test, $p < 0.001$) $n = 15$. No difference in performance is found among 30–120 V (*post-hoc* pairwise comparisons $p > 0.05$) $n = 16$–30. Further parametric behavioral analyses for aversive conditioning are shown in *Figure 1—figure supplement 1*. Bars and error bars represent mean and SEM, respectively.

The following figure supplement is available for figure 1:

**Figure supplement 1**. Conditioning with electric shock induces significant aversive visual memory.

We next analyzed the temporal requirements for neurons labeled in *TH-GAL4* and *DDC-GAL4* in our learning paradigm. We measured visual memories for 30 min retention and transiently blocked the neurons either during training (*Figure 2B*) when reinforcers were presented or during retrieval of the memory (*Figure 2C*). The blockade with *DDC-GAL4* during training severely impaired appetitive memory, whereas the same blockade after the training did not significantly affect memory (*Figure 2D,E*). Similarly, the neurons labeled in *TH-GAL4* were required specifically during acquisition of aversive memory (*Figure 2F,G*). These results suggest that the neurons differentially labeled with *DDC-GAL4* and *TH-GAL4* mediate the formation of appetitive and aversive visual memories, likely acting as

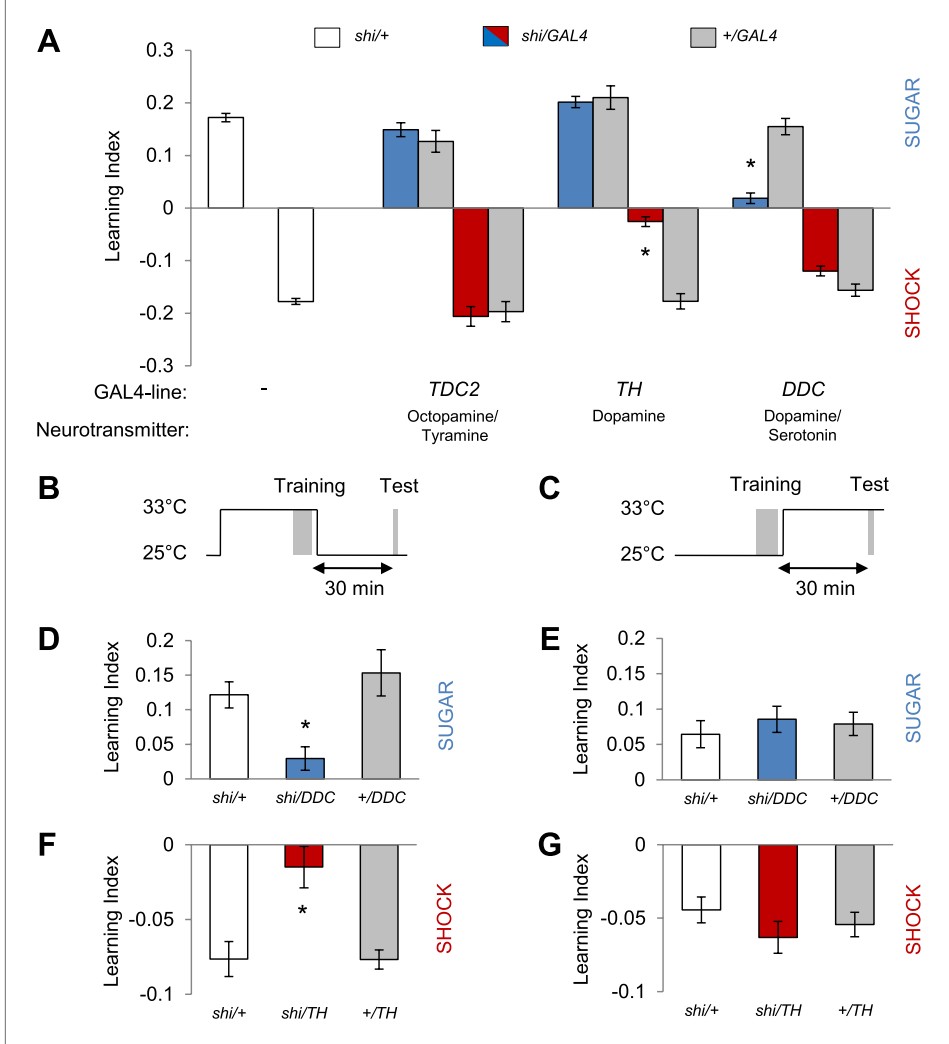

**Figure 2**. Different dopamine neurons are required for appetitive and aversive memory acquisition. (**A**) Different aminergic neurons are continuously blocked with corresponding GAL4 drivers. The blockade with *TH-GAL4* and *DDC-GAL4* selectively impaired aversive (one-way ANOVA, *post-hoc* pairwise comparisons, p < 0.001) and appetitive memories (Kruskal–Wallis test, *post-hoc* pairwise comparisons, p < 0.001), respectively. Blocking octopamine and tyramine neurons does not significantly impair memory (*post hoc* pairwise comparisons p > 0.05). n = 8–45. (**B** and **C**) Scheme of the temperature shift to block the output of corresponding neurons during training (**B**) or test (**C**). (**D** and **E**) Output of *DDC-GAL4* labeled neurons is only necessary in appetitive training (one-way ANOVA, *post hoc* pairwise comparisons, p < 0.05) but dispensable during test (one-way ANOVA, p > 0.05). n = 13–38. (**F** and **G**) Similarly, output of *TH-GAL4* labeled neurons is only necessary during aversive training (Kruskal–Wallis test, *post hoc* pairwise comparisons, p < 0.01) but dispensable during test (one-way ANOVA, p > 0.05). n = 12–16. All flies were starved prior to the experiments. Memory of the experimental group is compared to performances of the corresponding control groups. Only the most conservative statistical result of multiple pairwise comparisons is stated. Bars and error bars represent mean and SEM, respectively.

reinforcement signals. As specific subsets of dopamine neurons in *TH-GAL4* and *DDC-GAL4* have been shown to signal sugar reward and shock punishment for olfactory memories (***Claridge-Chang et al., 2009***; ***Aso et al., 2012***; ***Burke et al., 2012***; ***Liu et al., 2012***), we genetically dissected these populations further to identify the essential neurons for visual memories.

To functionally restrict the neurons in *DDC-GAL4* and *TH-GAL4* into smaller subsets, we selected two specific GAL4 driver lines for dopamine neurons: *MB504B* and *R58E02*. *R58E02-GAL4* drives GAL4 expression in the PAM cluster neurons that signal reward for olfactory memory (***Figure 3A,C***; ***Liu, et al., 2012***). This driver co-expresses with *DDC-GAL4*, but rarely with *TH-GAL4,* in the PAM cluster

**Table 1.** Sugar and shock response of the lines with impaired visual memories

| Genotype | Sugar response Mean ± SEM | Shock response Mean ± SEM | * p < 0.05 | Control data for |
|---|---|---|---|---|
| shi/+ | – | −0.429 ± 0.056 | | *Figures 2 and 3* |
| shi/TH | – | −0.189 ± 0.019 | * | *Figure 2* |
| +/TH | – | −0.386 ± 0.060 | | *Figure 2* |
| shi/MB504B | – | −0.382 ± 0.041 | | *Figure 3* |
| +/MB504B | – | −0.385 ± 0.051 | | *Figure 3* |
| shi/+ | 0.718 ± 0.020 | – | | *Figure 2* |
| shi/DDC | 0.729 ± 0.027 | – | | *Figure 2* |
| +/DDC | 0.725 ± 0.017 | – | | *Figure 2* |
| shi/+ | 0.436 ± 0.053 | – | | *Figure 3* |
| shi/R58E02 | 0.544 ± 0.033 | – | | *Figure 3* |
| +/R58E02 | 0.563 ± 0.036 | – | | *Figure 3* |
| CS | 0.547 ± 0.041 | −0.471 ± 0.044 | | *Figure 4* |
| dumb$^2$ | 0.634 ± 0.042 | −0.289 ± 0.031 | * | *Figure 4* |
| dumb$^2$/MB247, dumb$^2$ | – | −0.255 ± 0.028 | * | *Figure 4* |
| +/MB247 | – | −0.452 ± 0.067 | | *Figure 4* |
| shi/+ | 0.569 ± 0.039 † | −0.359 ± 0.034 | | *Figure 5* |
| shi/201y | 0.444 ± 0.039 | −0.401 ± 0.033 | | *Figure 5* |
| +/201y | 0.649 ± 0.064 | −0.353 ± 0.033 | | *Figure 5* |
| shi/+ | 0.569 ± 0.039 † | −0.387 ± 0.029 | | *Figure 5* |
| shi/MB247 | 0.535 ± 0.018 | −0.297 ± 0.038 | | *Figure 5* |
| +/MB247 | 0.575 ± 0.048 | −0.384 ± 0.028 | | *Figure 5* |
| shi/+ | 0.548 ± 0.024 | −0.366 ± 0.031 | | *Figure 7* |
| shi/MB010B | 0.526 ± 0.061 | −0.393 ± 0.052 | | *Figure 7* |
| +/MB010B | 0.591 ± 0.048 | −0.353 ± 0.040 | | *Figure 7* |
| shi/MB009B | 0.689 ± 0.067 | −0.368 ± 0.028 | | *Figure 7* |
| +/MB009B | 0.739 ± 0.017 | −0.373 ± 0.035 | | *Figure 7* |

No significant defect in naïve sugar preference is detected among the experimental groups and the corresponding control groups (one-way ANOVA, p > 0.05), n = 4–8. No significant defect in naïve shock avoidance is detected among the experimental groups and the corresponding control groups (one-way ANOVA, p > 0.05), except for shi/TH, dumb2 and dumb2/MB247, dumb2 (one-way ANOVA, *post-hoc* pairwise comparisons, p < 0.05). n = 6–10. Consistent with a study by **Lebestky et al. (2009)**, we observe prolonged arousal after shocking these flies, and this hyperactivity rather than shock sensitivity is a likely cause of the reduced avoidance. Indeed, the *DopR+* expression in the MB rescues visual memories, even though shock avoidance is still not intact.
†The identical data is represented, as these two sets of measurements were performed in parallel.
*indicates p-value lower than 0.05. Corrected p-values for shock avoidance: shi/TH vs. shi/+: p = 0.006, shi/TH vs. +/TH: p = 0.037; CS vs. dumb$^2$: p = 0.01, dumb$^2$/MB247,dumb$^2$ vs. CS/MB247: p = 0.0492

(**Liu et al., 2012**). *MB504B* is a Split GAL4 line we generated to specifically label four individual dopamine neurons in the PPL1 cluster: MB-MP1, MB-MV1, MB-V1, and the neuron that projects to the tip of the α lobe (**Figure 3B**). These neurons are a subset of *TH-GAL4* and have been shown, using a less specific line, to induce aversive olfactory memory (**Aso et al., 2012**). We found that the blockade of these neurons with *shi$^{ts1}$* indeed impaired aversive or appetitive visual memory, respectively (**Figure 3C–F**), but did not significantly affect the reflexive choice of sugar and shock (**Table 1**). Thus, we conclude that visual and olfactory memories share neuronal substrates for appetitive and aversive reinforcements.

To examine whether the activity of these neurons directly drives memories, or carries a regulatory role, we exerted direct control over neuronal activity with *R58E02-GAL4* and *MB504B-GAL4* using a

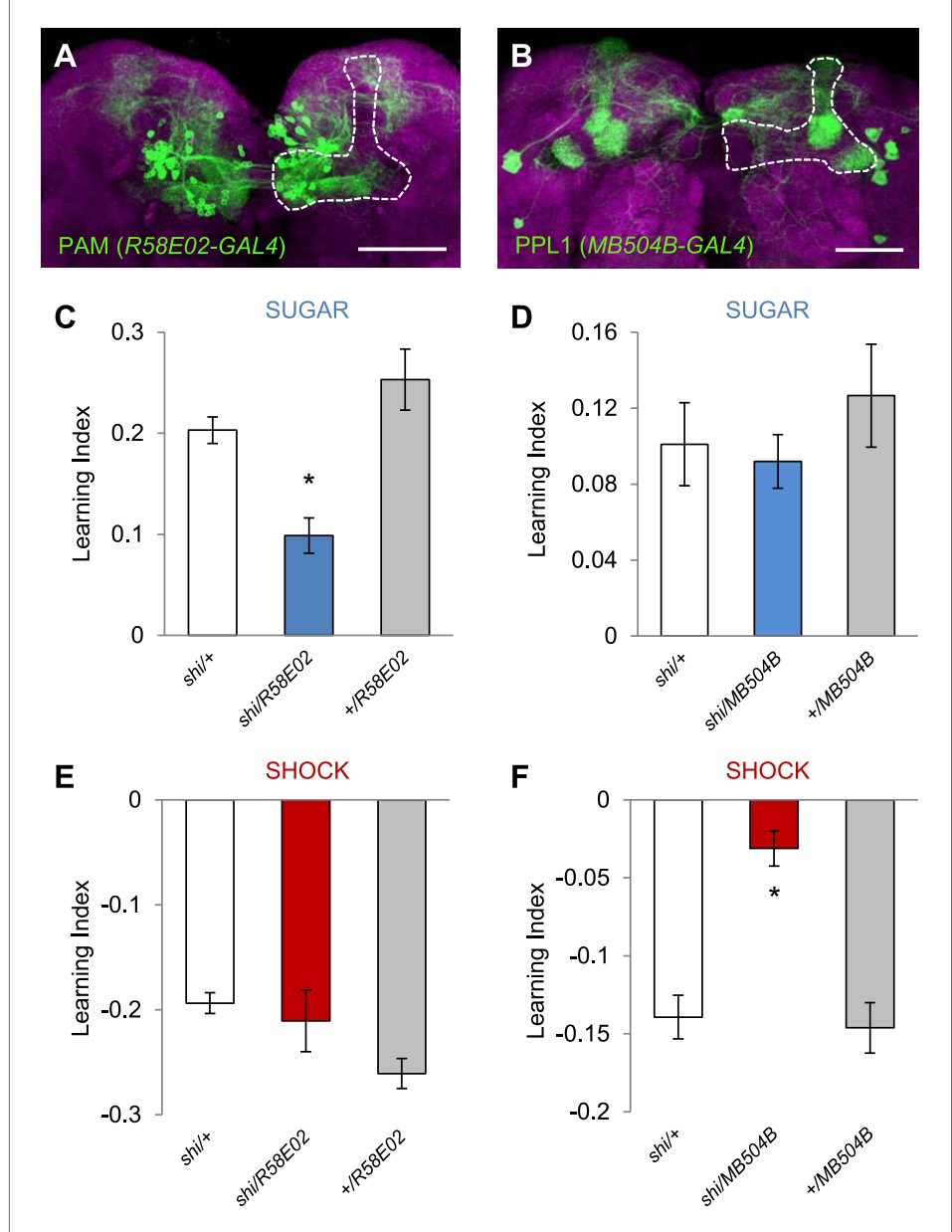

**Figure 3**. Different sets of dopamine neurons projecting to the MB are necessary for appetitive and aversive memories. (**A** and **B**) Expression patterns of *R58E02-GAL4* (PAM-Cluster) and *MB504B-GAL4* (PPL1-Cluster) in the MB region (outlined) are visualized by mCD8::GFP (green). Neuropil counterstaining with antibody against Synapsin (magenta). Scalebar = 50 μm. (**C** and **D**) Blocking *R58E02-GAL4* (**C**), but not *MB504B-GAL4* (**D**) subsets of dopamine neurons impairs appetitive memories, respectively (one-way ANOVA, *post-hoc* pairwise comparisons, $p < 0.001$). $n = 10–21$. (**E** and **F**) Blocking *MB504B-GAL4* (**E**), but not *R58E02-GAL4* (**F**) subsets of dopamine neurons impairs aversive memories, respectively (one-way ANOVA, *post-hoc* pairwise comparisons, $p < 0.001$). $n = 11–21$. All flies were starved prior to the experiments. Bars and error bars represent mean and SEM, respectively.

temperature-sensitive cation channel dTRPA1 (*Hamada et al., 2008*). We paired one of the visual stimuli with thermo-activation of GAL4-expressing neurons by raising ambient temperature to 31°C and subsequently measured the flies' color preference (*Figure 4A*). Thermo-activation of the PAM and PPL1 cluster neurons with *R58E02-GAL4* and *MB504B-GAL4* was sufficient to induce appetitive and aversive memories, respectively (*Figure 4B,C*). Based on these results we conclude that different subsets of dopamine neurons supply appetitive and aversive reinforcement information for visual as well as olfactory memories.

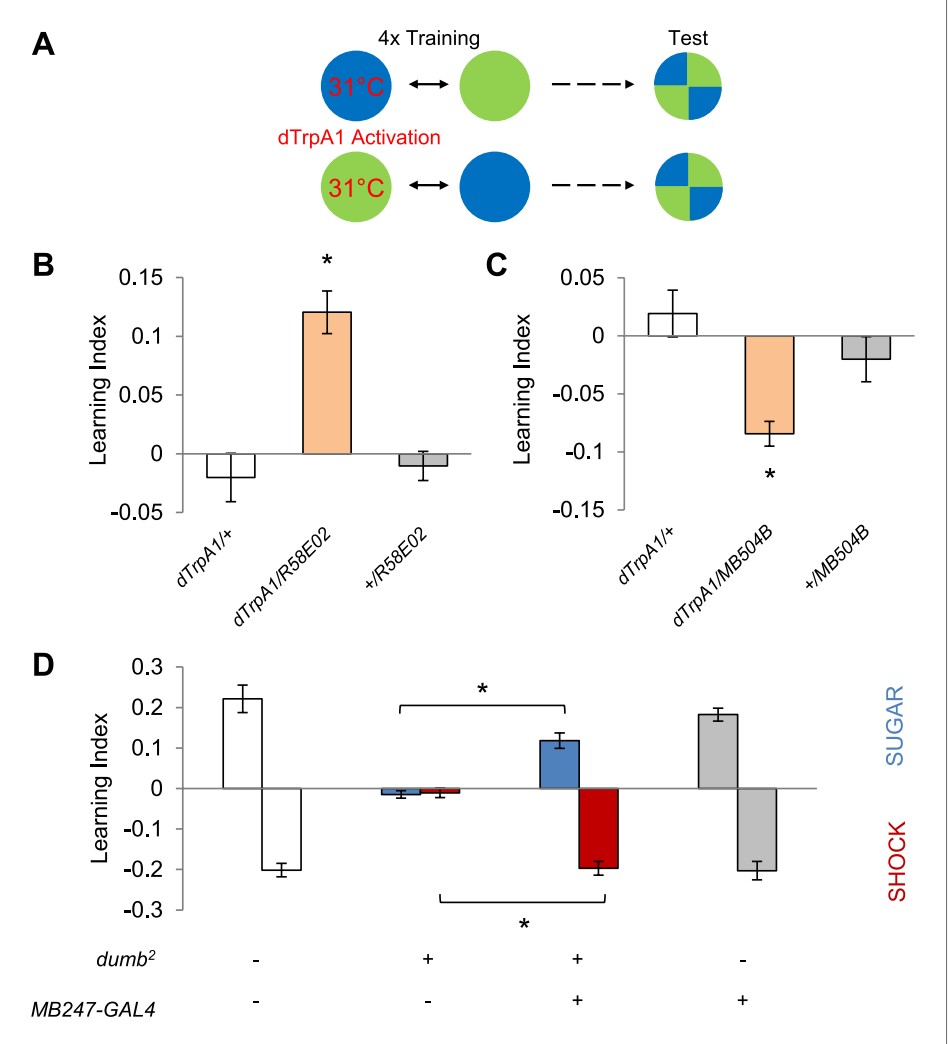

**Figure 4**. Different sets of dopamine neurons projecting to the MB are sufficient for appetitive and aversive memories. (**A**) Scheme of reinforcement replacement. One visual stimulus is paired with temperature elevation (31°C) during training, leading to activation of *dTrpA1*-expressing neurons. (**B** and **C**) Thermo-activation with *R58E02-GAL4* (PAM) and *MB504B-GAL4* (PPL1) induces appetitive (one-way ANOVA, *post-hoc* pairwise comparisons test, $p < 0.01$) and aversive visual memories (Kruskal–Wallis test, *post-hoc* pairwise comparisons test, $p < 0.05$), respectively. $n = 6–18$. (**D**) DopR null mutant *dumb²* (which also allows *dumb* expression via GAL4) shows a strong defect in appetitive and aversive memory (Kruskal–Wallis test, *post-hoc* pairwise comparisons test, $p < 0.001$). Expression of DopR⁺ in the MB restores both forms of visual memory of the *dumb²* mutant (Kruskal–Wallis test, *post-hoc* pairwise comparisons test, $p < 0.05$). $n = 8–16$. Visual cue discrimination for *dumb²* mutant flies is shown in *Figure 4—figure supplement 1*. All flies were starved prior to the experiments. Bars and error bars represent mean and SEM, respectively.

The following figure supplement is available for figure 4:

**Figure supplement 1**. Visual stimulus preference in the memory test after aversive conditioning.

Since DopR, a D1-like dopamine receptor, is required for olfactory memories (*Kim et al., 2007*; *Liu et al., 2012*; *Qin et al., 2012*), we hypothesized that it is also required for visual memories. Consistent with our results from the block of dopamine neurons, we found severe appetitive and aversive visual memory defects in the mutant for DopR (*dumb²*; *Kim et al., 2007*, *Figure 4D*, see *Table 1*; *Figure 4—figure supplement 1* for controls). As both the PAM neurons in *R58E02-GAL4* and the PPL1 neurons in *MB504B-GAL4* terminate in the MBs (*Figure 3A,B*), we hypothesized that their output is transmitted to MB intrinsic neurons, Kenyon cells (KCs), through DopR. To express DopR⁺ in the mutant background,

we made use of the PiggyBac insertion mutant (*dumb²*) that contains UAS in the first intron of the *DopR* gene allowing GAL4-dependent expression of the gene (*Kim et al., 2007*). Selective expression of *DopR⁺* in the KCs using *MB247-GAL4* significantly rescued the memory defect of the mutant (*Figure 4D*). Altogether, these results suggest that the same sets of dopamine neurons convey reward and punishment signals to the MBs to induce appetitive and aversive memories of the different sensory modalities.

## Kenyon cells are required for visual memories

If visual information is modulated by converging dopamine signals in the MBs, the output of KCs should be essential for visual memories. To test this hypothesis, we used two distinct GAL4 drivers labeling α/β and γ KCs, *201y* (*Yang et al., 1995*) and *MB247* (*Zars et al., 2000*), to express *shiᵗˢ¹* and continuously block the output of KCs during training and test. Both appetitive and aversive memories in the experimental groups were significantly impaired (*Figure 5A*, see *Table 1* and *Figure 5—figure supplement 1* for controls). To control for expression of *shiᵗˢ¹* outside the MBs, we blocked GAL4 transactivation of *201y* in

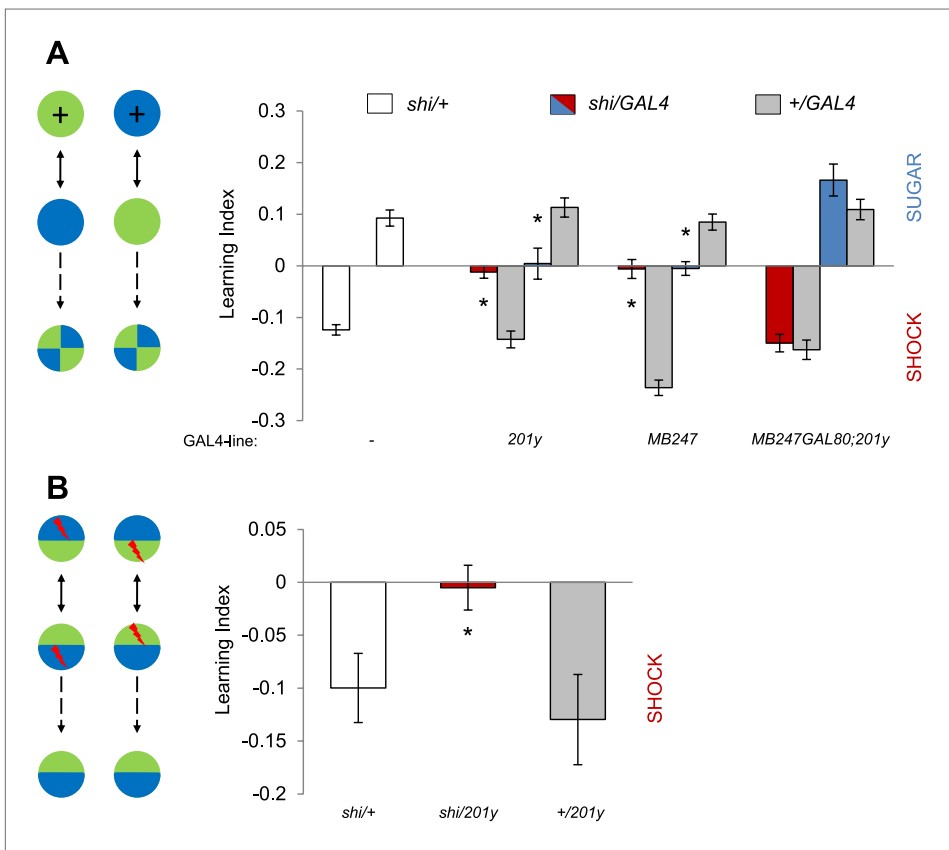

**Figure 5**. MBs are necessary for visual memories. (**A**) Blocking output of KCs labeled with *201y-GAL4* and *MB247-GAL4* leads to significant impairment in both appetitive and aversive memories (one-way ANOVA, *post-hoc* pairwise comparisons, $p < 0.05$). $n = 10$–14. *MB247-GAL80* restores impaired memory with *201y-GAL4* (*post-hoc* pairwise comparisons, $p > 0.05$). $n = 10$–14. Visual cue discrimination of these genotypes is shown in *Figure 5—figure supplement 1*. (**B**) Conditioning protocol with operant component and with visual context maintained between training and test (compare to protocol in **A**). Visual memories with the modified protocol require intact MBs (one-way ANOVA, *post-hoc* pairwise comparisons, $p < 0.05$). $n = 14$–15. *Figure 5—figure supplement 2* shows another example for the requirement of MBs with a modified aversive conditioning protocol. Bars and error bars represent mean and SEM, respectively.

The following figure supplements are available for figure 5:

**Figure supplement 1**. Visual stimulus preference in the memory test after aversive conditioning.

**Figure supplement 2**. Shock before test is dispensable at high temperature, however, requirement of MBs remains.

the MBs using *MB247-GAL80* (**Krashes et al., 2007**). Addition of *MB247-GAL80* fully restored the impaired memories (**Figure 5A**). Thus, we conclude that visual memories require the output of KCs.

The MBs have been reported to be dispensable for some forms of visual memory, especially in the 'flight simulator' (**Wolf et al., 1998**; **Ofstad et al., 2011**) and to be required only when the learning context is changed between training and test (**Liu et al., 1999**; **Peng et al., 2007**). Our conditioning design also involves a change in the context of visual stimulation: the entire conditioning arena is homogeneously illuminated during training, whereas green and blue lights are presented in the four quadrants of the arena in the test (**Figure 5A**). To eliminate this context change, we modified the conditioning design by simultaneously presenting both visual cues throughout training and test (**Figure 5B**). This necessarily introduced an 'operant' component to the training similar to that of the standard flight simulator learning; flies can avoid electric shock by staying away from the paired color. *201y-GAL4* flies with blocked MBs displayed no visual memory, whereas the control flies significantly avoided the punished color (**Figure 5B**). Thus we conclude that visual memory in our assay requires the MBs independent of the conditioning design (classical vs operant) and of context changes between training and test. Also the additional arousal of flies prior to the test by a single pulse of electric shock did not change the requirement for MB output (**Figure 5—figure supplement 2**).

## Distinct but overlapping subsets of Kenyon cells are required for visual and olfactory memories

Visual and olfactory memories use the same MB-projecting dopamine neurons. We therefore asked whether the post-synaptic MB neurons are also shared, using *c305-GAL4* (**Krashes et al., 2007**), *17D-GAL4* (**Martin et al., 1998**) and *201y-GAL4* to inactivate selective KC subsets during aversive visual and olfactory conditioning. Blocking the α'/β' neurons with *c305a-GAL4* selectively impaired olfactory memory (**Figure 6A,B**). In contrast, the consequences of the blockades with *201y-GAL4* (α/β and γ neurons) and *17D-GAL4* (α/β neurons) were the same in visual and olfactory memories. *17D-GAL4/shi^{ts1}* flies had no significant perturbation of either memory, while the blockade of the α/β and γ neurons with *201y-GAL4* strongly impaired both visual and olfactory memories (**Figure 6A,B**). Hence, visual and olfactory memories recruit partially overlapping KC subsets. The specific contribution of α'/β' neurons to olfactory learning is consistent with the preferential representation of olfactory inputs to these neurons (**Turner et al., 2008**).

## The γ lobe neurons are important for visual memory

In olfactory memories, the different lobes of the MBs have specific functions. To map the contributions of MB lobes to visual learning, we took a suite of Split GAL4 drivers that specifically label different lobes (**Figure 7A**, 'Materials and methods' section). Blocking the output of the γ lobe neurons (**Figures 7A**,

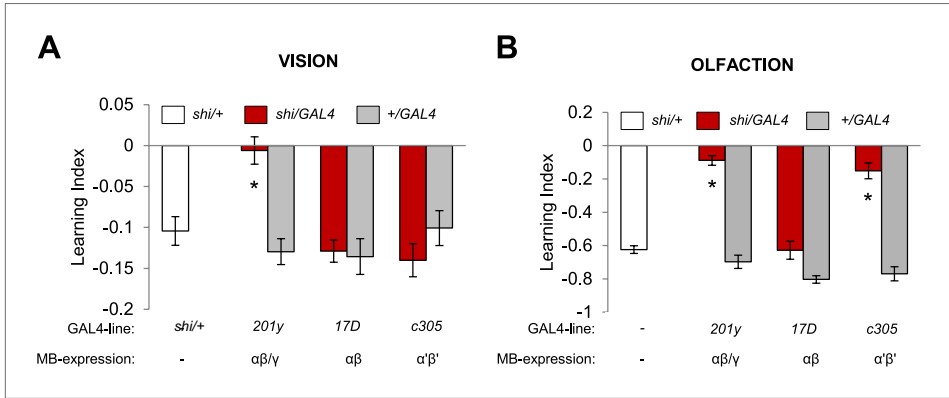

**Figure 6**. Overlapping, yet distinct sets of MB-lobes are needed for aversive visual and olfactory learning. (**A**) Only blocking output of neurons labeled with *201y-GAL4* (α/β and γ lobes) during visual conditioning leads to memory impairment (one-way ANOVA, *post-hoc* pairwise comparison, p < 0.01), *n* = 12–19. (**B**) In contrast, in aversive olfactory memory, blocking output of neurons labeled with *201y-GAL4* (α/β and γ lobes) and *c305-GAL4* (α'/β' lobes) leads to significant memory impairment (one-way ANOVA, *post-hoc* pairwise comparison, p < 0.001). *n* = 10–22. Bars and error bars represent mean and SEM, respectively.

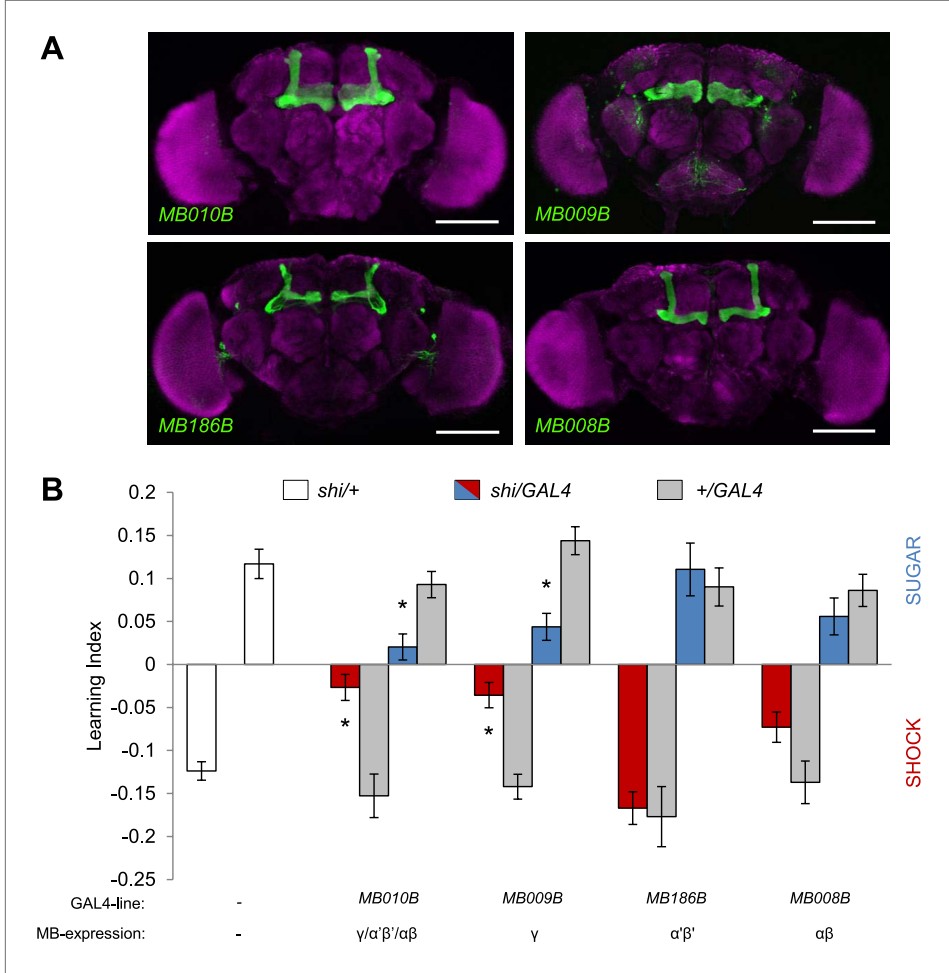

**Figure 7**. MB γ lobes are required for visual memory. (**A**) Partial projections of expression patterns of Split GAL4 lines are visualized by mCD8::GFP (green). Neuropil counterstaining with antibody against Synapsin (magenta). Scalebar = 100 μm. *MB010B-GAL4* labels all lobes of the MB, *MB009B-GAL4* labels γ lobes, *MB186B-GAL4* labels α′/β′ lobes, *MB008B-GAL4* labels α/β lobes. (**B**) Blocking output of specific MB-lobe subsets during appetitive conditioning showed that γ lobes are specifically required (Kruskal–Wallis test, *post-hoc* pairwise comparisons, p < 0.05). n = 10–23. In aversive conditioning we found similar requirement of the γ lobes (one-way ANOVA, *post-hoc* pairwise comparisons, p < 0.05). n = 8–23. Visual cue discrimination of these genotypes is shown in ***Figure 7—figure supplement 1***. Bars and error bars represent mean and SEM, respectively.

The following figure supplement is available for figure 7:

**Figure supplement 1**. Visual stimulus preference in the memory test after aversive conditioning.

*MB009B-GAL4*, see ***Table 1*** and ***Figure 7—figure supplement 1*** for controls) as well as the entire KC population (***Figures 7A***, *MB010B-GAL4*, see ***Table 1*** and ***Figure 7—figure supplement 1*** for controls) impaired both appetitive and aversive visual memories (***Figure 7B***).

Blocking the output of α′/β′ neurons (***Figures 7A***, *MB186B-GAL4*) or α/β neurons (***Figures 7A***, *MB008B-GAL4*) did not significantly reduce the performance compared to the controls (***Figure 7B***), although *MB008B-GAL4/UAS-shi^{ts1}* flies showed a tendency toward impairment. The required KC subsets for appetitive and aversive memories were strikingly similar. This suggests that the output of specific subsets of KCs representing visual information is differentially modulated by reward or punishment.

## MB output is required during training and test

Next, we explored the temporal requirements of MB output for the formation and retrieval of visual memory. We transiently blocked the output of a broad population of KCs using *MB010B-GAL4*

(*Figure 8A–D*, all lobes) (*Bräcker et al., 2013*) and *MB247-GAL4* (*Figure 8A,B,E,F*; α/β and γ lobes) and found requirement for KCs in memory acquisition and retrieval. Interestingly, the transiently blocked output of a smaller set of KCs using *201y-GAL4* (α/β and γ lobes) *and MB009B-GAL4* (γ lobes) revealed a selective requirement for the retrieval but not the formation of aversive visual memory (*Figure 8A,B,G,J*). Output of KCs labeled by *MB247-GAL4* (α/β and γ lobes) was also required for acquisition and retrieval of appetitive memory (*Figure 8A,B,K,L*). We thus conclude that different KCs mediate acquisition and retrieval of visual memories.

## Discussion

### High-throughput aversive visual conditioning

Devising a transparent electric shock grid module made it possible to apply the same visual stimulation in aversive and appetitive conditioning assays. We also developed an integrated platform for fully automated high-throughput data acquisition using customized software to control the presentation of electric shock and visual stimuli while making video recordings of behavior (*Figure 1*; *Schnaitmann et al., 2010*, *2013*). In our assays, memory performance is based on altered visual preference in walking flies, a task likely to be less demanding than the constant flight required for flight simulator learning. These advantages facilitate behavioral examination of many genotypes.

### Associative memories of different modalities share mushroom body circuits

Circuits underlying olfactory and visual memory can be optimally compared when the sugar reward and electric shock punishment are matched between the two modalities. We found that visual and olfactory memories share the same subsets of dopamine neurons that convey reinforcing signals (*Figures 2,3 and 4*). This shared requirement of the transmitter system between visual and olfactory learning has been described in crickets (*Unoki et al., 2005*, *2006*; *Mizunami et al., 2009*). However, the pharmacological manipulation used in these studies does not allow further circuit dissection.

For electric shock reinforcement, identified neurons in the PPL1 cluster, such as MB-MP1, MB-MV1 and MB-V1, drive aversive memories in both visual and olfactory learning (*Figures 3F and 4C*; *Claridge-Chang et al., 2009*; *Aso et al., 2010*, *2012*), while the MB-M3 neurons in the PAM cluster seem to be involved specifically in aversive olfactory memory (*Figure 2A*, data not shown) (*Aso et al., 2010*, *2012*). Thus, overlapping sets of dopamine neurons appear to represent electric shock punishment in both visual and olfactory learning with olfactory aversive memory probably recruiting a larger set. We previously showed that the MB-M3 neurons induce aversive olfactory memory that increases stability of other memory components (*Aso et al., 2012*). Olfactory memories last longer than visual memories (data not shown; *Schnaitmann et al., 2010*) potentially due to the recruitment of additional dopamine neurons.

In appetitive conditioning, PAM cluster neurons play crucial roles in both olfactory and visual memories (*Figure 2A,D,E*, *Figure 3C,E*, *Figure 4B*; *Burke et al., 2012*; *Liu et al., 2012*). Which cell types in these clusters are involved and whether there is a cellular distinction between olfactory and visual memory requires further analysis at the single cell level. Most importantly, all these neurons convey dopamine signals to restricted subdomains of the MB. The blockade of octopamine neurons did not impair appetitive visual memories with sucrose (*Figure 2A*). The involvement of octopamine neurons may be more substantial when non-nutritious sweet taste rewards are used, as has been shown in olfactory learning (*Burke et al., 2012*).

In addition to these shared reinforcement circuits in the MB, the necessity of MB output for visual memory acquisition and retrieval is also consistent with olfactory conditioning (*Figure 8*; *Dubnau et al., 2001*; *McGuire et al., 2001*; *Schwaerzel et al., 2002*; *Krashes et al., 2007*), although the recruited KC subsets are not identical for different modalities (*Figure 6*). Taken together, these results suggest that the MBs harbor associative plasticity for visual memories and support the conclusion that similar coincidence detection mechanisms are used to form memories within the MBs (*Figure 9*) (*Heisenberg, 2003*; *Gerber et al., 2004b*; *Qin et al., 2012*). Centralization of similar brain functions spares the cost of maintaining similar circuit motifs in different brain areas and may be an evolutionary conserved design of information processing. Such converging inputs of different stimuli into a multisensory area have even been described in humans (*Beauchamp et al., 2008*).

'Flight simulator' visual learning was shown to require the central complex but not the MBs (*Wolf et al., 1998*; *Liu et al., 2006*; *Pan et al., 2009*). Although this appears to contradict our study, we note

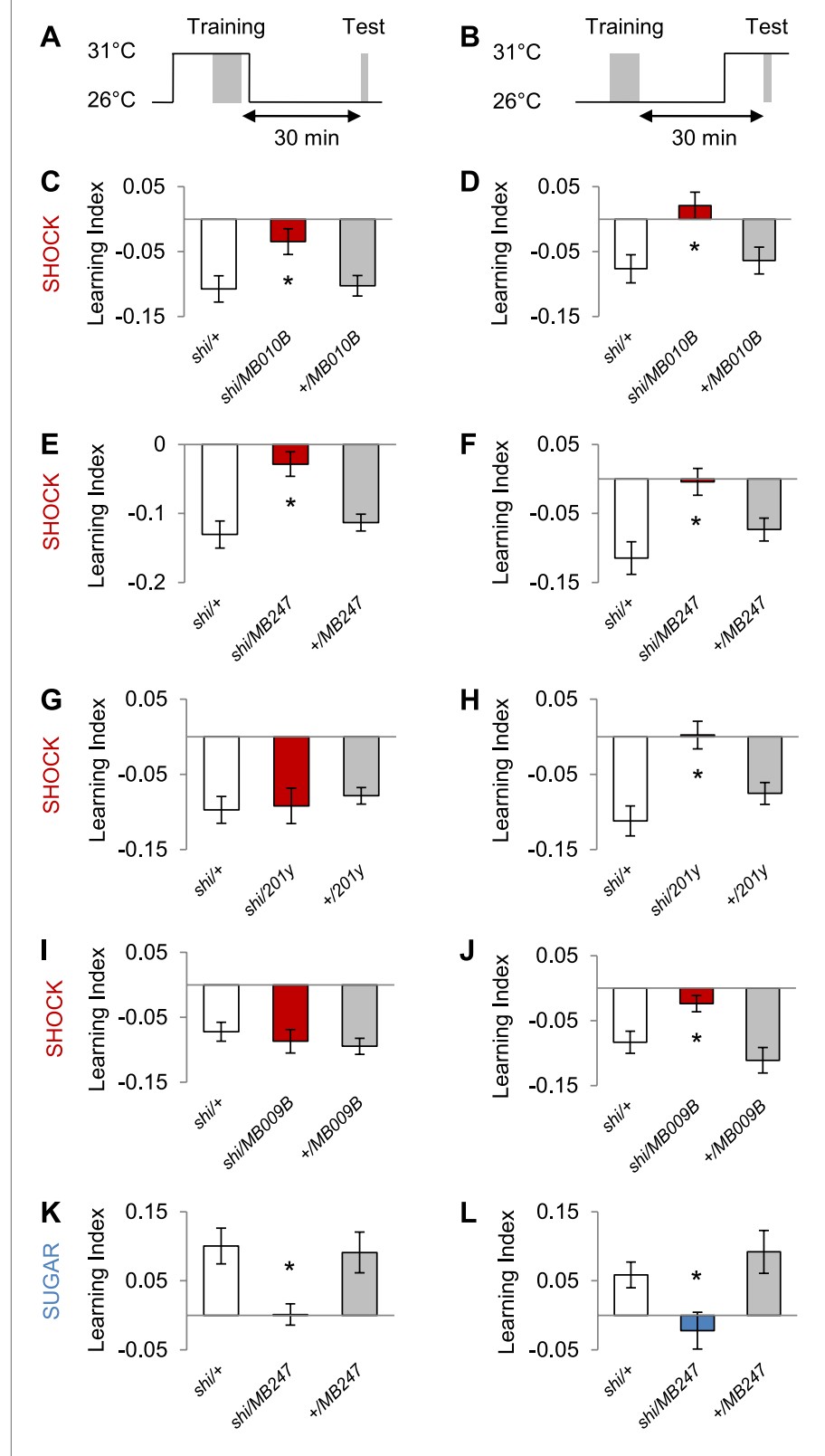

**Figure 8**. MB output is needed during visual memory acquisition and retrieval. (**A** and **B**) Scheme of the temperature shift to block the output of corresponding neurons during training (**A**) or test (**B**). (**C** and **D**) Output of neurons labeled with *MB010B-GAL4* is necessary during aversive training and test (one-way ANOVA, *post-hoc* pairwise

*Figure 8. Continued*

comparisons, p < 0.05), *n* = 7–13. (**E** and **F**) Output of neurons labeled with *MB247-GAL4* is necessary during aversive training and test (one-way ANOVA, *post-hoc* pairwise comparison, p < 0.05). *n* = 10–16. (**G** and **H**) Output of neurons labeled with *201y-GAL4* is dispensable during training (one-way ANOVA, p > 0.05), *n* = 20–22, but necessary during aversive test (one-way ANOVA, *post-hoc* pairwise comparisons, p < 0.01) *n* = 12–17. (**I** and **J**) Output of neurons labeled with *MB009B-GAL4* is dispensable during training (one-way ANOVA, p > 0.1), *n* = 8, but necessary during aversive test (one-way ANOVA, *post-hoc* pairwise comparisons, p < 0.05) *n* = 8. (**K** and **L**) Output of *MB247*-labeled neurons is needed during appetitive training and test (one-way ANOVA, *post-hoc* pairwise comparison, p < 0.05). *n* = 10–28. Bars and error bars represent mean and SEM, respectively.

that there are important differences between the behavioral paradigms employed. In the flight simulator, a single tethered flying *Drosophila* is trained to associate a specific visual cue with a laser beam punishment, to later on avoid flying towards this cue in the test. Although we controlled for visual context consistency and the 'operant component' of the flight simulator training, any other difference could account for the differential requirement of brain structures. Given that flies during flight show octopamine-mediated modulation of neurons in the optic lobe (*Suver et al., 2012*), similar state-dependent mechanisms might underlie different requirement of higher brain centers. Thus, it is critical to design comparable memory paradigms.

## Differentiated sensory representations in the mushroom body?

This study together with the results in associative taste learning (*Masek and Scott, 2010*; *Keene and Masek, 2012*) highlights the fact that the role of the MB in associative learning is not restricted to one sensory modality or reinforcer (*Figure 9*). We found that olfactory and visual memories recruit overlapping, yet partly distinct, sets of Kenyon cells (*Figures 6,9*). In contrast to the well-described olfactory projection neurons, visual inputs to the MB remain unidentified. No anatomical evidence has been reported in *Drosophila* for direct connections between optic lobes and MBs (*Otsuna and Ito, 2006*; *Mu et al., 2012*) although such connections are found in other insects (*Mobbs, 1982*; *Schildberger, 1984*; *Li and Strausfeld, 1997*; *Gronenberg and Hölldobler, 1999*; *Paulk and Gronenberg, 2008*; *Lin and Strausfeld, 2012*). Also afferents originating in the protocerebrum were found to provide multi-modal input to the MB lobes of cockroaches (*Li and Strausfeld, 1997*). Thus, *Drosophila* MBs may receive indirect visual input from optic lobes, and

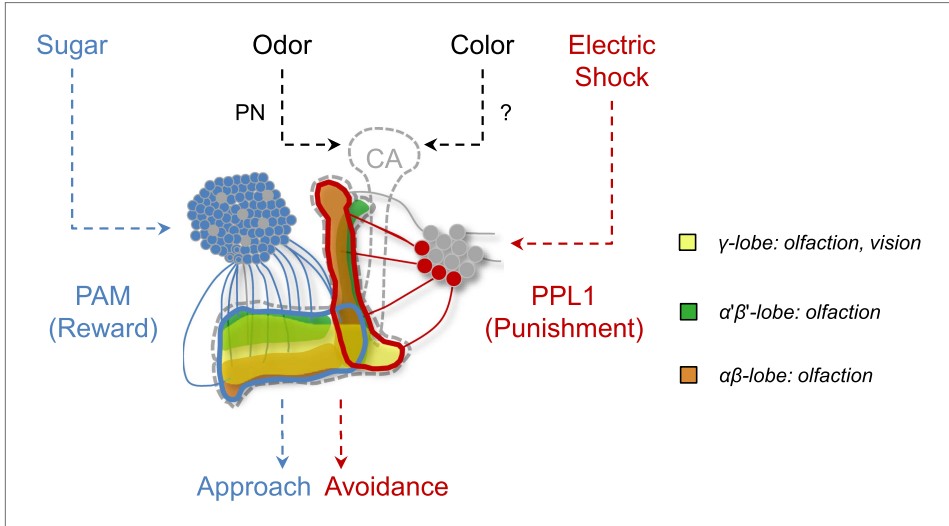

**Figure 9**. Circuit model of olfactory and visual short-term memories. Visual and olfactory information is conveyed to partially overlapping sets (γ lobe neurons) of KCs. Olfactory input to the calyx (CA) via projection neurons (PN) is well characterized, whereas the visual input to the MB has not been identified yet. Output of KCs, representing olfactory and visual information is locally modulated by the different subsets of dopamine neurons (PAM, PPL1) to form appetitive and aversive memories.

the identification of such a visual pathway would significantly contribute to our understanding of the MB circuits.

Given the general requirement of the γ lobe neurons (*Figure 7*), visual and olfactory cues may be both represented in the γ neurons. Consistently, the dopamine neurons that convey appetitive and aversive memories heavily project to the γ lobe (*Figure 3A,B*; *Claridge-Chang et al., 2009*; *Aso et al., 2010*, *2012*; *Burke et al., 2012*; *Liu et al., 2012*). In olfactory conditioning, the γ lobe was shown to contribute mainly to short-term memory (*Zars et al., 2000*; *Isabel et al., 2004*; *Blum et al., 2009*; *Trannoy et al., 2011*; *Qin et al., 2012*). This converging evidence from olfactory and visual memories suggests a general role for the γ lobe in short-lasting memories across different sensory modalities (*Figures 6,7 and 9*). Previous studies found that the MB is also involved in sensorimotor gating of visual stimuli or visual selective attention (*van Swinderen and Greenspan, 2003*; *Xi et al., 2008*; *van Swinderen et al., 2009*). Therefore, the MB circuits for visual associative memories might be required for sensorimotor gating and attention.

Interestingly, the contribution of the α'/β' lobes is selective for olfactory memories (*Figure 6*; *Krashes et al., 2007*; *Cervantes-Sandoval et al., 2013*). This Kenyon cell class is more specialized to odor representation, as the cells have the broadest odor tuning and the lowest response threshold among the three Kenyon cell types (*Turner et al., 2008*).

The role of α/β neurons in visual memories is also limited (*Figures 6A and 7B*). The α/β neurons might play more modulatory roles in specific visual memory tasks, such as context generalization, facilitation of operant learning and occasion setting (*Liu et al., 1999*; *Tang and Guo, 2001*; *Brembs and Wiener, 2006*; *Brembs, 2009*; *Zhang et al., 2013*). This modulatory role of the α/β neurons is corroborated in olfactory learning, where they are preferentially required for long-lasting memories (*McGuire et al., 2003*; *Isabel et al., 2004*; *Blum et al., 2009*; *Trannoy et al., 2011*; *Keleman et al., 2012*; *Xie et al., 2013*).

Differentiated but overlapping sensory representations by KCs may be conserved among insect species. In honeybees, different sensory modalities are represented in spatially segregated areas of the calyx, whereas the basal ring region receives visual and olfactory inputs (*Mobbs, 1982*; *Strausfeld, 2002*). The MB might thus have evolved to represent the sensory space of those modalities that are subject to associative modulation.

## Materials and methods

### Flies and genetic crosses

Flies were reared at 25°C, at 60% relative humidity under a 12–12 hr light–dark cycle on a standard cornmeal-based food. As all transgenes were inserted into the $w^-$ mutant genome, the X chromosomes of strains were replaced with that of wild-type Canton-S ($w^+$). We used $F_1$ progenies of crosses between females of genotypes *UAS-dTrpA1* (*Hamada et al., 2008*), *UAS-shi^ts* (*Kitamoto, 2001*), *MB247-GAL80;UAS-shi^ts* (*Krashes et al., 2007*), *UAS-mCD8::GFP* (*Lee and Luo, 1999*) or WT-females and males of genotypes *TH-GAL4* (*Friggi-Grelin et al., 2003*), *DDC-GAL4* (*Li et al., 2000*), *R58E02-GAL4* (*Liu et al., 2012*), *TDC2-GAL4* (*Cole et al., 2005*), *MB247-GAL4* (*Zars et al., 2000*), *c305a-GAL4* (*Krashes et al., 2007*), *17D-GAL4* (*Martin et al., 1998*), *201y-GAL4* (*Yang et al., 1995*), or Canton-S males. The expression patterns of drivers for KCs were compared previously (*Aso et al., 2009*). The *dumb^2* null mutant was used to localize the cells that receive dopamine signals (*Kim et al., 2007*).

To identify a role for dopamine neurons and specific lobes of the MB in visual learning we utilized specific Split GAL4 lines. Split GAL4 lines have high specificity in expression pattern, since here the DNA-binding domain (DBD) and the activation domain (AD) of the GAL4-protein were independently targeted by different promoters. In this way, the UAS transgene is only expressed where the expression patterns of the two enhancers intersect and therefore the functional GAL4-protein can be reconstituted (*Luan et al., 2006*; *Pfeiffer et al., 2010*). We used $F_1$ progenies of crosses between females of genotypes *UAS-shi^ts* (*Kitamoto, 2001*), *UAS-mCD8::GFP* (*Lee and Luo, 1999*), *UAS-dTrpA1* (*Hamada et al., 2008*) or WT-females and males of genotypes *MB504B-GAL4*, *MB010B-GAL4* (*Bräcker et al., 2013*), *MB009B-GAL4*, *MB008B-GAL4*, *MB186B-GAL4* (*Bräcker et al., 2013*) or Canton-S males. Split GAL4 lines were generated using the vectors described in *Pfeiffer et al. (2010)* by inserting *R52H03-p65ADZp* into attp40 and *TH-ZpGAL4DBD* into attP2 (*MB504B-GAL4*), *R13F02-p65ADZp* into attP40 and *R52H09-ZpGAL4DBD* into attP2 (*MB010B-GAL4*), *R13F02-p65ADZp* into attP40 and *R45H04-ZpGAL4DBD* into attP2 (*MB009B-GAL4*), *R13F02-p65ADZp* into attP40 and *R44E04-ZpGAL4DBD* into attP2 (*MB008B-GAL4*) and *R52H09-p65ADZp* into attP40 and *R34A03-ZpGAL4DBD* into attP2 (*MB186B-GAL4*). Detailed

methods for generating and evaluating MB Split GAL4 driver lines will be described elsewhere (Aso et al., in preparation).

As we were unable to distinguish genotype or sex in our behavioral videos, we sorted flies by genotype under $CO_2$ anesthesia at least 2 days prior to experiments. Hence, for appetitive conditioning experiments 2–4 day old flies were starved in moistened empty vials to approximately 20% mortality (*Schnaitmann et al., 2010*). For aversive conditioning, starvation was not applied unless otherwise explicitly stated. Behavioral experiments each used 30–40 mixed males and females under dim red light in a custom-made plastic box, containing a heating element on the bottom and a fan for air circulation.

## Apparatus for appetitive conditioning and visual stimulation

Our appetitive conditioning paradigm was as previously described (*Schnaitmann et al., 2010*, *2013*), except we used LED (instead of an LCD screen) to present visual stimuli (green and blue light) from beneath the fly (*Figure 1A,B*) (*Schnaitmann et al., 2013*). We constructed a stimulation module using computer-controlled high-power LEDs with peak wavelengths 452 nm and 520 nm (Seoul Z-Power RGB LED, Korea) or 456 nm and 520 nm (H-HP803NB, and H-HP803PG, 3W Hexagon Power LEDs, Roithner Lasertechnik, Austria) for blue and green stimulation, respectively. LEDs were housed in a base (144 mm below the arena), which allowed homogeneous illumination of a filter paper as a screen (*Figure 1D*) (*Schnaitmann et al., 2013*). Using a custom-made software and controlling device we were able to illuminate four quadrants of the arena independently when required (*Schnaitmann et al., 2013*). For separate illumination of each quadrant, the light paths of LEDs were separated by light-tight walls in a cylinder with air ducts (*Figure 1A–C*). The intensities were controlled by current and calibrated using a luminance meter BM-9 (Topcon Technohouse Corporation, Japan) or a PR-655 SpectraScan Spectroradiometer, Chatsworth, CA, USA,: 14.1 $Cd/m^2$ s (blue) and 70.7 $Cd/m^2$ s (green) (*Schnaitmann et al., 2013*). Each quadrant was equipped with an IR-LED (850 nm), that was used for background illumination, for example, during the preference/avoidance test.

## Apparatus for aversive conditioning

For aversive electric shock conditioning, we developed a new apparatus module containing an arena with a transparent shock grid (*Figure 1E*). The arena itself consisted of the transparent shock grid on the bottom, a plastic ring as a wall and a glass lid. The shock grid was a custom-made ITO-coated glass plate (9 × 9 cm; Diamond Coatings Ltd., UK). ITO is a conductive transparent substance. A grid was laser-etched onto the ITO glass in order to insulate the positive and negative electrodes (lanes in the grid were 1.6 mm spaced 0.1 mm apart, Lasermicronics GmbH, Germany). We applied alternating current. The two halves of the grid can be independently controlled. The plastic ring (wall) and the glass lid were coated with diluted Fluon (10%; Fluon GP1, Whitford Plastics Ltd., UK) to prevent flies from walking on the lid and wall. Consequently, flies were forced to stay on the shock grid on the bottom of the arena. A filter paper was clamped underneath the shock grid and served as a screen.

A trigger (Universal-Impulsgenerator UPG 100, ELV Elektronik AG, Leer, Germany) for activating the custom-made electric shock generator was controlled by the same custom-made software used to control the LEDs (*Schnaitmann et al., 2013*). During the test phase, the shock arena was video recorded from above with a CMOS camera (Firefly MV, Point Grey, Richmond, Canada) controlled by custom-made software (*Schnaitmann et al., 2010*). Four setups were run in parallel.

## Behavioral protocols for appetitive and aversive learning

We used equivalent experimental designs for appetitive and aversive conditioning; in each, differential conditioning was followed by binary choice without reinforcement (*Figure 1*, *Figure 5A*; *Schnaitmann et al., 2010*). Briefly, in a single conditioning experiment, approximately 40 flies were introduced into the arena using an aspirator. During a training trial, the whole arena was illuminated with alternating green and blue light (60 s each; conditioned stimuli = CS), one of the colors was paired with reinforcement (unconditioned stimulus = US).

For appetitive conditioning, filter paper soaked in high concentration (2 M) of sucrose and dried was presented as a reward (*Schnaitmann et al., 2010*). For aversive conditioning, one second of electric shock (AC 60 V) was applied 12 times in 60 s during CS+ (CS paired with reinforcement) presentation. The consecutive CS+ and CS− presentations were interspersed with 12 s intervals without illumination (*Schnaitmann et al., 2010*). Training trials were repeated four times per experiment if not otherwise stated (*Figure 1—figure supplement 1A*).

In the test, administered 60 s after the end of the last training session, flies were allowed to choose between blue and green, which were each presented in two diagonally opposite quadrants of the arena (unless otherwise stated). The distribution of the flies was video recorded for 90 s at 1 frame per second (*Schnaitmann et al., 2010*). No US was presented in the test period. For aversive conditioning, a 1 s shock pulse (90 V) was applied 5 s before the beginning of the test to arouse the flies (*Figure 1— figure supplement 1B*). However when testing flies at high temperature (33°C) this additional shock was dispensable for aversive memory retrieval (*Figure 5—figure supplement 2*). Two groups trained with reciprocal CS–US pairing (Green+/Blue− and Blue+/Green−) were trained in the same setup consecutively. The difference in visual stimulus preference between the two groups was then used to calculate a learning index for each video frame (*Schnaitmann et al., 2010*). Reinforcement was paired with the first visual stimulus in half of the experiments, and with the second in the remaining experiments, to cancel any effect of order (*Schnaitmann et al., 2010*).

Control responses to sugar and shock were measured as described previously (*Schnaitmann et al., 2010*). The arenas used for appetitive and aversive conditioning were backlit with IR-LEDs, and flies were given a choice between two halves of the arena, one with the US presented as in the training and one without US. Their behavior was recorded for 60 s using the same video setup. A preference index was calculated by subtracting the numbers of flies on the US half from the numbers on the control half, divided by the total number of flies.

By use of a heating element and fan we were able to raise the temperature around the apparatus to a constant 33°C. In temperature shift experiments flies were transferred into moistened empty vials and kept in darkness, while the temperature was adjusted from permissive (25/26°C) to restrictive (31/33°C) or vice versa. The test was performed 30–40 min after training and started 60 s after reintroduction of the flies.

## Reinforcement substitution with thermo-activation by TrpA1

We established a new behavioral protocol for reinforcement substitution for visual memories using *dTrpA1* expression as in olfactory conditioning (*Aso et al., 2010*). Flies were 'trained' as for conditioning, but the conditioned visual stimulus was paired not with sugar or shock but with high temperature that leads to thermo-activation of dTRPA1-expressing neurons (*Figure 4A*). Flies were transferred between two plastic vials with different visual stimuli (blue and green as for conditioning) and different temperatures (24°C and 31°C). 5 s before the onset of the visual stimulation, flies were gently tapped into the vial on the corresponding apparatus. The two CS presentations in each training trial were intermitted by a 60 s interval at 24°C. After four training trials, flies were kept at 24°C for 60 s in the transfer vial and 60 s in the dark test apparatus before testing at 24°C. Control flies not expressing *dTrpA1* and wild-type flies did not show conditioned visual preference (*Figure 4B,C*). Significant memory in this assay is thus driven by appetitive or aversive reinforcement signals from thermo-activation.

## Aversive conditioning without context change

In contrast to the standard classical conditioning protocol, CS+ and CS− were simultaneously presented in the two halves of the arena during training and test (*Figure 5B*). The half of the arena displaying CS+ was electrified. Flies were allowed to choose between the differently cued two halves for 20 s, and this training trial was repeated eight times with an inter-trial interval of 20 s without stimulus. The sides of CS/US presentation were pseudo-randomized to avoid potential prediction of the next side to be shocked. During the test, the two halves were illuminated as in training, but without shock, creating a shared context between training and test.

## Olfactory conditioning

Standard olfactory conditioning was applied (*Tully and Quinn, 1985*; *Schwaerzel et al., 2002*). Differential conditioning with two odors (3-octanol and benzaldehyde) followed by binary choice without reinforcement. Each odor presentation lasted 60 s, and one of the odors was paired with 12 pulses of electric shock (100 V DC). Immediately after training, flies were tested for memory performance by measuring conditioned odor avoidance in the T-maze.

## Statistics

Statistical analyses were performed with Prism5 software (GraphPad). Groups that did not violate the assumption of normal distribution (Shapiro–Wilk test) and homogeneity of variance (Bartlett's test) were analyzed with parametric statistics: one-sample *t*-test or one-way analysis of variance followed by

the planned pairwise multiple comparisons (Bonferroni). Experiments with data that were significantly different from the assumptions above were analyzed with non-parametric tests, such as Mann–Whitney test or Kruskal–Wallis test followed by Dunn's multiple pair-wise comparison. The significance level of statistical tests was set to 0.05. Only the most conservative statistical result of multiple pairwise comparisons is indicated.

## Immunohistochemistry

Adult fly brains were dissected, fixed and stained using standard protocols (*Aso et al., 2009*). Synapsin antibody (*Klagges et al., 1996*) combined with Cy3-conjugated goat anti-mouse antibody were used to visualize the neuropil. Anti-GFP antibody was used to increase the intensity of the GFP signal (rabbit polyclonal to GFP [Invitrogen] with Alexa Fluor488-conjugated goat anti-rabbit as the secondary antibody). Frontal optical sections of whole-mount brains were sampled with a confocal microscope (Olympus FV1000). Images of the confocal stacks were analyzed with the open-source software Fiji (*Schindelin et al., 2012*).

## Acknowledgements

We are thankful to Anja B Friedrich for providing confocal data for *R58E02-GAL4* and Igor Siwanowicz for the setup sketch.

## Additional information

### Funding

| Funder | Grant reference number | Author |
|---|---|---|
| Bundesministerium für Bildung und Forschung | Bernstein Focus Neurobiology of Learning 01GQ0932 | Hiromu Tanimoto |
| Deutsche Forschungsgemeinschaft | TA552/5-1 | Hiromu Tanimoto |
| Ministry of Education, Culture, Sports, Science, and Technology/Japan Society for the Promotion of Science | KAKENHI 25890003 | Hiromu Tanimoto |
| Max-Planck-Gesellschaft | | Hiromu Tanimoto |
| Howard Hughes Medical Institute | | Yoshinori Aso, Gerald M Rubin |
| Boehringer Ingelheim Fonds | | Christopher Schnaitmann |
| Ministry of Education, Culture, Sports, Science, and Technology/Japan Society for the Promotion of Science | KAKENHI 26120705 | Hiromu Tanimoto |
| Ministry of Education, Culture, Sports, Science, and Technology/Japan Society for the Promotion of Science | KAKENHI 26250001 | Hiromu Tanimoto |
| Ministry of Education, Culture, Sports, Science, and Technology/Japan Society for the Promotion of Science | KAKENHI 26119503 | Hiromu Tanimoto |

The funders had no role in study design, data collection and interpretation, or the decision to submit the work for publication.

### Author contributions

KV, CS, Conception and design, Acquisition of data, Analysis and interpretation of data, Drafting or revising the article; KVD, SK, Acquisition of data, Analysis and interpretation of data, Drafting or revising the article; YA, GMR, Drafting or revising the article, Contributed unpublished essential data or reagents; HT, Conception and design, Analysis and interpretation of data, Drafting or revising the article

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
