## [Decision Letter]

Thank you for sending your work entitled “Shared mushroom body circuits underlie visual and olfactory memories in *Drosophila*” for consideration at *eLife.* Your article has been favorably evaluated by a Senior editor, a Reviewing editor (Mani Ramaswami) and 2 reviewers.

The Reviewing editor and the two reviewers discussed their comments before we reached this decision, and the Reviewing editor has assembled the following comments to help you prepare a revised submission.

By developing behavioral assays for aversive and appetitive visual learning, the authors compare the circuitry required for visual learning with that previously established for olfactory learning. The question is interesting and assessed with the extremely good reagents and technologies expected from the authors. The conclusions mentioned in the Abstract describing dopaminergic neurons and mushroom body neurons required for visual learning, as well as how they compare with those required for olfactory learning, are interesting, significant, and deserve to be published in *eLife*.

1) One consensus experimental requirement is to provide clear information on the specificity of PAM DA neurons are for sugar learning and the PPL1 DA neurons for shock learning. This will require (reciprocal type) experiments to show that PAM DA neurons are not required for visual forms of shock learning and that PPL1 DA neurons are not required for visual forms of sugar learning. The fact that specificity has been demonstrated for olfactory learning is appreciated, but this needs to be demonstrated for visual learning as well.

2) The authors should consider a deeper discussion of the comparative roles of gamma lobes in short-term olfactory and visual forms of learning. If the authors have data on whether the gamma lobes are required for acquisition and/or retrieval of visual memory, then it will be a useful addition to the paper.

3) The manuscript is well written for a select group of *Drosophila* neuroscientists, intimately familiar with learning. The manuscript text will benefit from being edited to make it appropriate for a broader readership of *eLife.*

---

## [Author Response]

*1) One consensus experimental requirement is to provide clear information on the specificity of PAM DA neurons are for sugar learning and the PPL1 DA neurons for shock learning. This will require (reciprocal type) experiments to show that PAM DA neurons are not required for visual forms of shock learning and that PPL1 DA neurons are not required for visual forms of sugar learning. The fact that specificity has been demonstrated for olfactory learning is appreciated, but this needs to be demonstrated for visual learning as well*.

We performed additional experiments and blocked different subsets of dopamine neurons with *MB504B-GAL4* and *R58E02-GAL4* and tested for appetitive and aversive memories, respectively. The dopamine neurons labeled in *R58E02-GAL4* are not required for aversive visual memory, and the dopamine neurons labeled in *MB504B-GAL4* are not required for appetitive visual memory either. Thus, we confirmed the differential requirement of the dopamine subsets in the PAM and PPL1 clusters for appetitive and aversive visual learning. We included these data in Figure 3. For practical reasons we split the original figure into two. Thus, selective requirement of dopamine neurons is now shown in Figure 3, whereas sufficiency of dopamine neurons is shown in new Figure 4.

*2) The authors should consider a deeper discussion of the comparative roles of gamma lobes in short-term olfactory and visual forms of learning. If the authors have data on whether the gamma lobes are required for acquisition and/or retrieval of visual memory, then it will be a useful addition to the paper*.

As the reviewers pointed out, the contribution of the gamma lobe for olfactory conditioning is mainly to short-term memory (30; 6; 77). Particularly, dopamine input to the gamma lobe through DopR is sufficient to restore the memory defect (63), and Rutabaga function (supposedly downstream of DopR) in the gamma lobes is sufficient to restore short-term memory (92). Our study showed for the first time that gamma lobes are also essential for visual memory. This converging evidence suggests the general role for the formation of short-term memory. We added this discussion in the revision.

Based on the reviewers’ suggestion we examined the temporal requirement of the gamma neurons by performing an additional set of experiments using *MB009B-GAL4*, which specifically labels the gamma neurons. We found that output of gamma lobe KCs is only required during test. This is consistent with the results obtained with *201y-GAL4*. Given that the outputs of the dopamine neurons heavily project to the gamma lobe and are only required during training, short-term visual memory is likely formed inside the MB gamma lobes. We now compiled all temperature shift experiments with MB drivers in Figure 8 for a better comparison between lines.

*3) The manuscript is well written for a select group of* Drosophila *neuroscientists, intimately familiar with learning. The manuscript text will benefit from being edited to make it appropriate for a broader readership of* eLife*.*

Thanks to this comment, we now include elaborated explanation and discussion about general information on associative memory, previous findings, different learning assays, and the contrast of memories in different sensory modalities.